# A relationship between Autism-Spectrum Quotient and face viewing behavior in 98 participants

**Kira Wegner-Clemens, Johannes Rennig, Michael S. Beauchamp***

Department of Neurosurgery and Core for Advanced MRI, Baylor College of Medicine, Houston, Texas, United States of America

* michael.beauchamp@bcm.edu

## Abstract

Faces are one of the most important stimuli that we encounter, but humans vary dramatically in their behavior when viewing a face: some individuals preferentially fixate the eyes, others fixate the mouth, and still others show an intermediate pattern. The determinants of these large individual differences are unknown. However, individuals with Autism Spectrum Disorder (ASD) spend less time fixating the eyes of a viewed face than controls, suggesting the hypothesis that autistic traits in healthy adults might explain individual differences in face viewing behavior. Autistic traits were measured in 98 healthy adults recruited from an academic setting using the Autism-Spectrum Quotient, a validated 50-statement questionnaire. Fixations were measured using a video-based eye tracker while participants viewed two different types of audiovisual movies: short videos of talker speaking single syllables and longer videos of talkers speaking sentences in a social context. For both types of movies, there was a positive correlation between Autism-Spectrum Quotient score and percent of time fixating the lower half of the face that explained from 4% to 10% of the variance in individual face viewing behavior. This effect suggests that in healthy adults, autistic traits are one of many factors that contribute to individual differences in face viewing behavior.

## Introduction

A propensity for individuals with Autism Spectrum Disorder (ASD) to avoid eye contact during social interactions was noted as early as 1943 [1]. The development of video-based computerized eye tracking has allowed for careful quantification of the patterns of eye fixations made by individuals with ASD when viewing faces. While there is substantial variability across individual studies, a comprehensive meta-analysis of 122 independent studies concluded that "individuals with ASD show a reliable pattern of gaze abnormalities that suggest a basic problem with selecting socially-relevant versus irrelevant information" [2]. Abnormalities in gaze when viewing faces and other socially relevant stimuli are important because they may underlie the difficulties with social communication and social interaction that are a core feature of ASD.

A typical pattern of results from a study comparing control participants with ASD individuals is shown in Fig 1A, reprinted from [3]. When ten control participants (with no history of

**Data Availability Statement:** Data are available from Dryad: https://doi.org/10.5061/dryad.zpc866t5c.

**Funding:** This research was funded by R01NS065395 to MSB. The funder had no role in study design, data collection and analysis, decision to publish, or preparation of the manuscript.

**Competing interests:** The authors have declared that no competing interests exist.

psychiatric or neurological disease and no family history of autism) viewed static emotional faces, fixations were concentrated around the eyes and nose of the viewed face, with fewer fixations to the mouth. In contrast, when ten participants with autism (defined using the diagnostic criteria of the DSM-IV/ICD-10, Autism Diagnostic Interview, and Autism Diagnostic Observation Schedule) viewed the same faces, fixations were concentrated around the mouth of the viewed face, with fewer fixations of the nose and eyes. A related study reported that both adult and adolescent participants with ASD fixate the mouth more than controls when watching speech scenes from movies [4].

In addition to different face-viewing behavior between control and ASD individuals, a growing literature demonstrates variability in face-looking behavior *within* control participants. For instance, in a study of eighteen undergraduate students by Mehoudar *et al.* [5], the average pattern of face fixations was distributed between the eyes, nose and mouth of the viewed face, matching the pattern shown in Fig 1A and the original description by Yarbus [6]. However, when Mehoudar *et al.* [5] examined individual participant fixation maps, a more complex picture emerged. Some individuals mainly fixated the eyes, others the mouth, and others the nose. Importantly, individuals' fixation patterns remained consistent across an 18-month interval, demonstrating that they reflect distinct face viewing strategies rather than random variability across testing sessions. Patterns of face looking that vary across individuals have been reported for many different types of face stimuli, including dynamic talking faces, and a variety of viewing conditions, including real-world viewing of faces assessed with wearable eye trackers [5,7–12]. Fig 1B illustrates variability within control participants viewing short videos of actors speaking single syllables. Participant 19 spent the majority of fixation time on the right eye and nose of the talker, while Participant 75 spent the majority of fixation time on the mouth of the talker.

Group-level differences in face fixation between ASD and control participants (Fig 1A) appear similar to individual-level differences in face fixation between control participants (Fig 1B). This raises the question of whether autistic tendencies could underlie both sets of differences. To explore this question, we used the Autism-Spectrum Quotient, a self-report survey designed to be taken by adults of at least normal intelligence in order to measure social and non-social traits that have been linked to ASD. Participants respond to 50 forced choice questions, either agreeing or disagreeing with statements about their social skill, attention switching, attention to detail, communication, and imagination. The AQ has been used extensively to study autistic traits in healthy populations (reviewed in [13]).

The hypothesis underlying our study is that autistic traits predispose individuals to fixate the mouth rather than the eyes of a viewed face. Individuals with ASD will have high levels of such tendencies, meaning that they will primarily fixate the mouth of a viewed face. However, even within healthy adults not diagnosed with ASD, there is also a range of autistic tendencies that can be measured using the AQ. Control participants with higher levels of autistic tendencies might be more likely to fixate the mouth than control participants with lower levels of autistic tendencies. We tested 98 participants to allow sufficient statistical power to accurately measure the effect size of the relationship between AQ and face viewing behavior. A previous study with 36 participants interacting face-to-face with an experimenter found a relationship between autistic traits and visual exploration during social interactions [14].

## Methods

### Design and participants

Participants provided written informed consent under an experimental protocol approved by the Committee for the Protection of Human Participants of the Baylor College of Medicine,

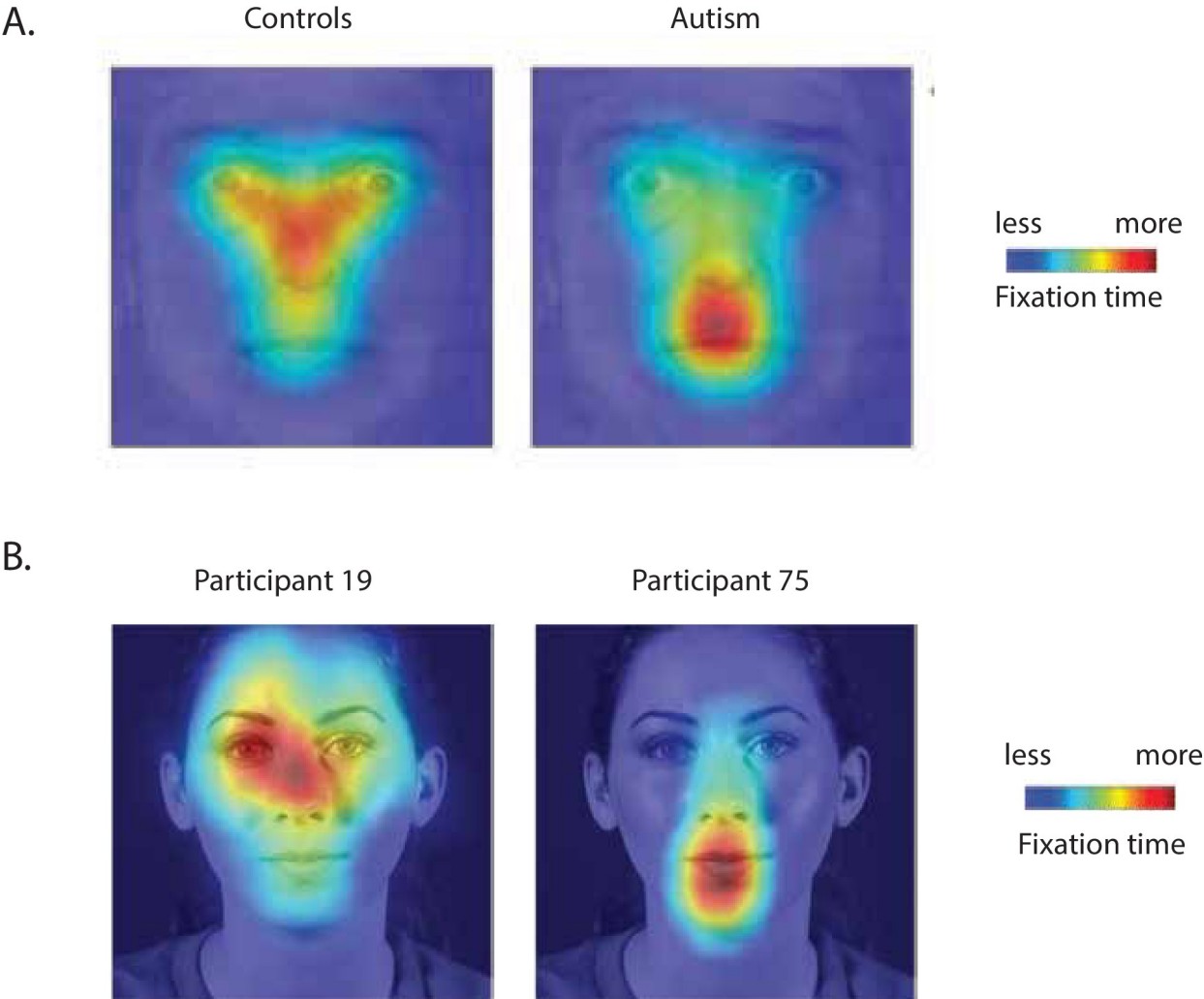

**Fig 1.** A. Fig 4 reproduced from (3). On the left, a fixation heat map averaged over group of 10 control participants is shown. On the right, a fixation heat map averaged over group of 10 ASD participants is shown. Warmer colors indicate more fixation time spent at that location; cooler colors indicate less time. B. Fixation heat maps for two individual participants in the current study viewing short movies of faces speaking syllables. The heat maps are visualized on a sample frame from one video, but represent data averaged across all frames and stimuli for that individual. Warmer colors indicate more fixation time spent at that location; cooler colors indicate less time.

Houston, TX. The individuals pictured in Figs 1, 2, 4 and 6 provided written informed consent (as outlined in the PLOS consent form) to publish their image alongside the manuscript. Data from the study have been deposited in the DataDryad repository and are available at https://doi.org/10.5061/dryad.zpc866t5c.

Participants were primarily recruited in a university setting, from Baylor College of Medicine and Rice University students and employees. A power analysis was conducted prior to collecting the data to determine sample size, based on an estimated expected effect size of r = 0.45 [14] which provided a suggested sample size of 71 (G*Power).

Participants were presented with stimuli consisting of auditory and visual recordings of human talkers and then completed the Autism-Spectrum Quotient as a measure of autistic traits. Stimuli were presented using Matlab with the Psychtoolbox extensions. In experiment 1, participants (n = 98; 66 female, mean age 21, age range 18–45) were shown short syllable videos recorded in lab and given an explicit task. In experiment 2, participants (n = 70, 49 female,

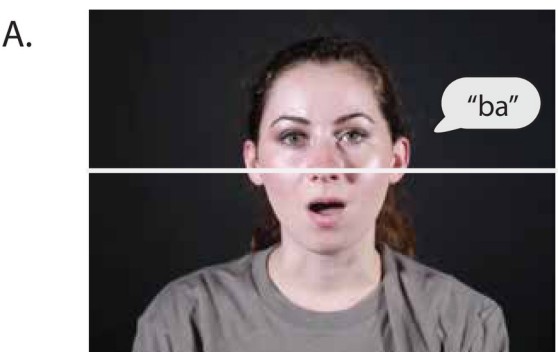

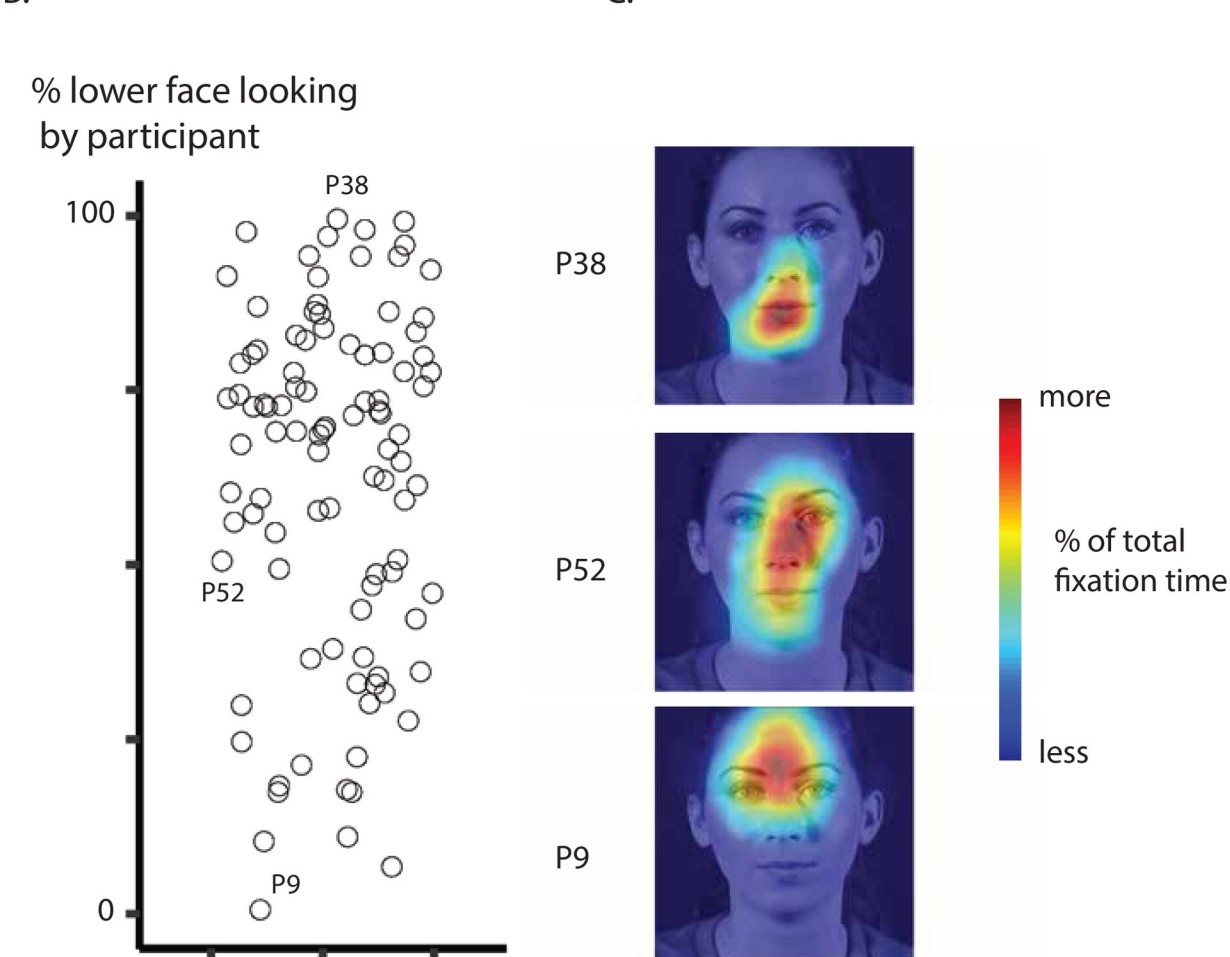

**Fig 2.** A. The stimuli in Experiment 1 consisted of 2 second movies of talkers speaking syllables. A single frame from one of the stimulus movies is shown; speech bubble indicated auditory speech component ("ba"), not present in actual stimulus movie. The white line (not present in actual stimulus movie) shows dividing line between upper and lower face regions of interest, used for analysis of eye fixation data. B. Face viewing

behavior during Experiment 1 summarized as percent of total fixation time spent fixating the lower face region of interest. A scatter plot of percent of total time spent fixating the lower face is shown, with value on the y axis and the x axis is jittered for visibility. Each symbol represents one participant. Symbols for three representative participants are labeled by participant number (fixation heat maps for labeled participants shown in *C*). C. Fixation heat maps for representative participants are shown (labeled by participant number; symbols representing each participant labeled in *B*). The heat maps are visualized on a sample frame from one video, but represent data averaged across all frames and stimuli. Warmer colors indicate more fixation time spent at that location; cooler colors indicate less time.

mean age 20, range 18–45; all native English speakers) were shown full sentence videos excerpted from longer videos without an explicit task.

To illustrate the different stimulus conditions and the resulting eye fixations, the figures show still frames from the stimulus videos. For videos recorded in the laboratory (Figs 1, 2 and 6) the talkers provided written informed consent permitting free distribution of the recordings, including in publications. For videos downloaded from YouTube (Figs 3 and 6), written informed consent could not be obtained so an image of first author KWC is shown instead.

## Autism-Spectrum Quotient

Participants completed the Autism-Spectrum Quotient (AQ) online before the test session [15]. This self-administered instrument consists of 50 items designed to assess autistic traits in

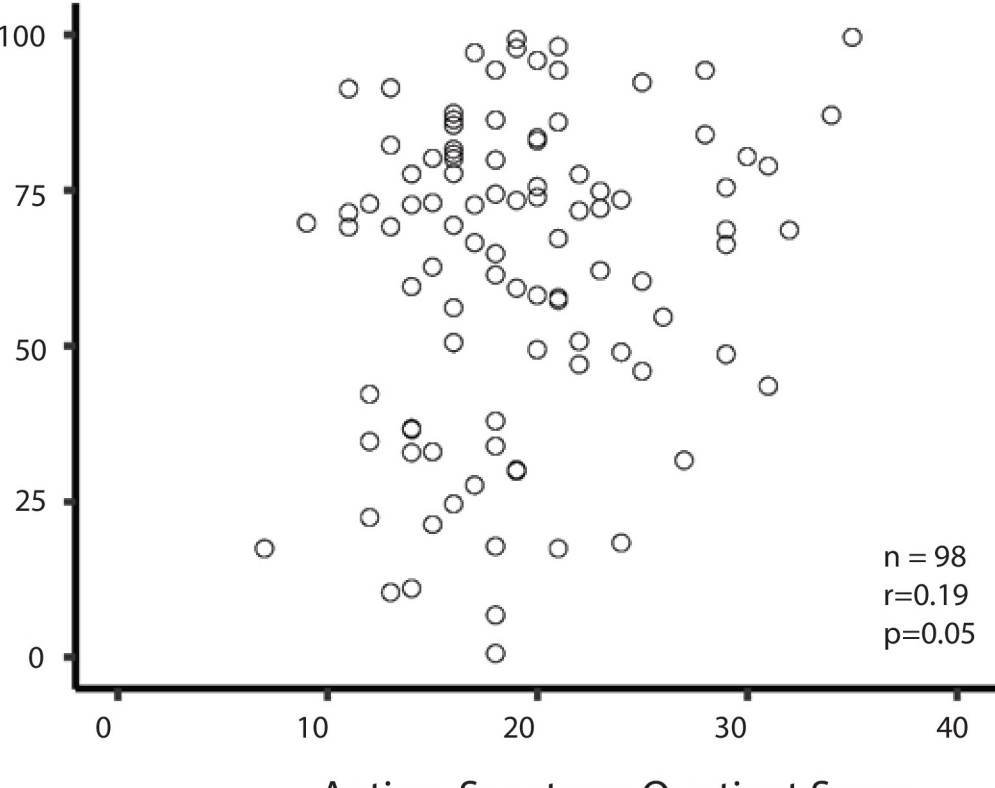

**Fig 3. The relationship between percent of time spent fixating the lower face while viewing syllable videos and Autism-Spectrum Quotient score.** Participant's percent lower face looking time is shown on the y axis; AQ score is shown on the x axis. Each symbol represents one participant.

A.

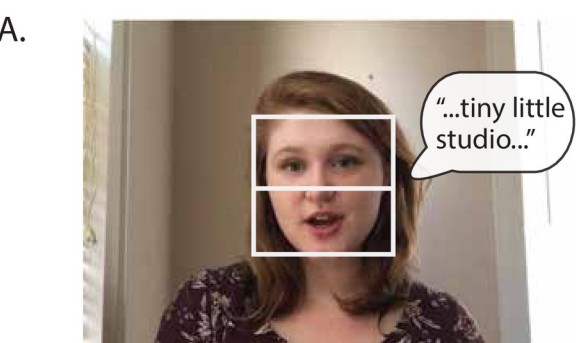

B.

## % lower face looking by participant

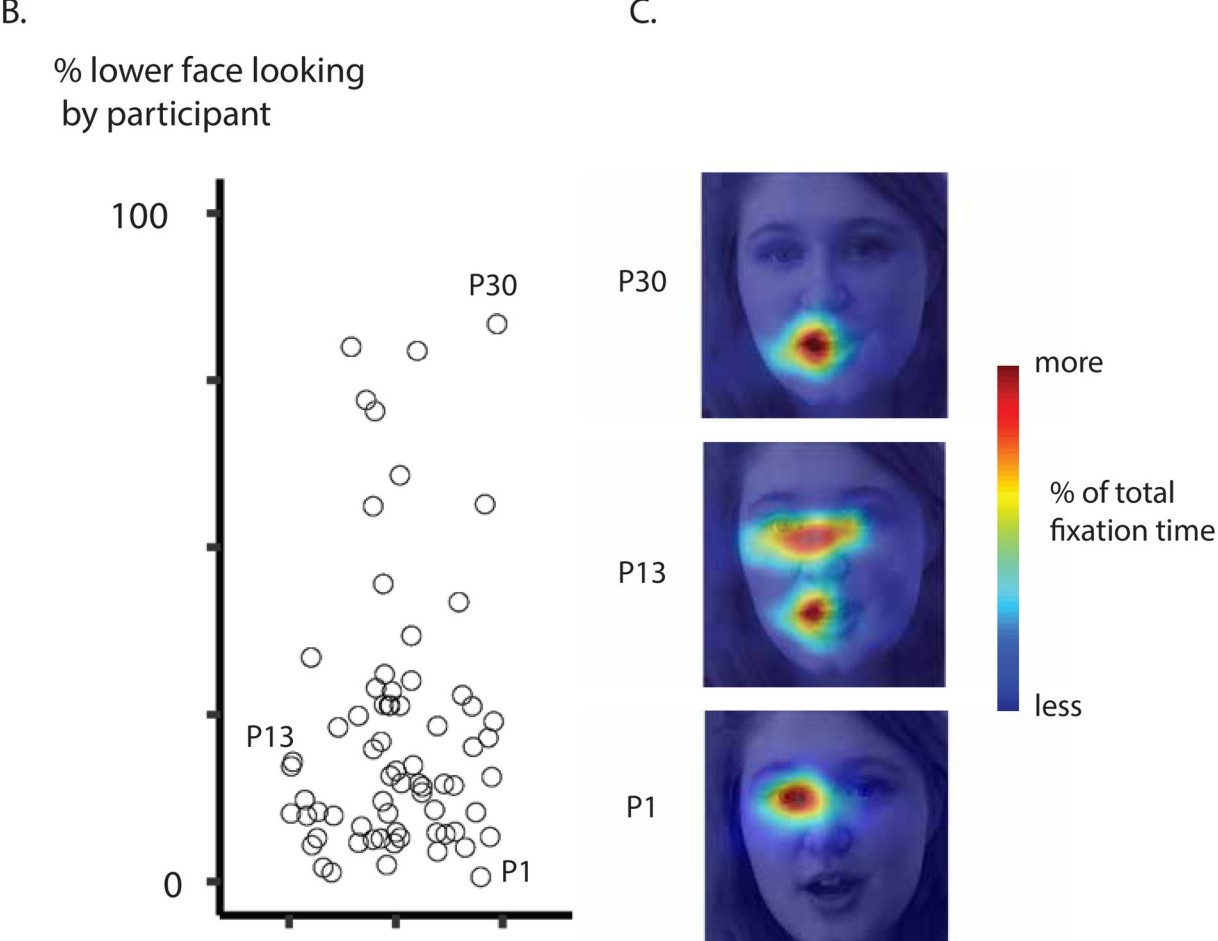

**Fig 4.** A. The stimuli in Experiment 2 consisted of movies between 13 and 42 seconds long of talkers speaking full sentences. An example frame is shown; speech bubble shows a portion of auditory speech component (" . . .tiny little studio . . ."), not present in actual stimulus movie. The white box (not present in actual stimulus movie) shows the region of data included and the white line (not present in actual stimulus movie)

shows the dividing line between upper and lower face regions of interest, used for analysis of eye fixation data. B. Face viewing behavior during Experiment 2 was summarized as percent of total fixation time spent fixating the lower face region of interest. A scatter plot of percent of total time spent fixating the lower face is shown, with value on the y axis and the x axis is jittered for visibility. Each point is a participant. Symbols for three representative participants are labeled by participant number (fixation heat maps for labeled participants shown in *C*). C. Fixation heat maps for representative participants are shown (labeled by participant number; symbols representing each participant labeled in *B*). The heat maps are visualized on a sample frame from one video, but represent data averaged across all frames and stimuli. Warmer colors indicate more fixation time spent at that location; cooler colors indicate less time.

adults of normal intelligence. The items reflect the differences in social skills, communication skills, imagination, attention to detail, and attention-switching noted in ASD. Each item is a statement about the participant's ability or preference, which the participant rates as definitely agree, slightly agree, slightly disagree, or definitely disagree. The test was scored using the method described in the original presentation of the test: each answer was collapsed into one of two categories (yes, for "definitely agree" or "slightly agree"; no, for slightly disagree or definitely disagree); assigned a value of 1 or 0, depending on the question; and the scores across all questions summed. A higher score indicates a higher degree of autistic-like traits, with a maximum score of 50 points.

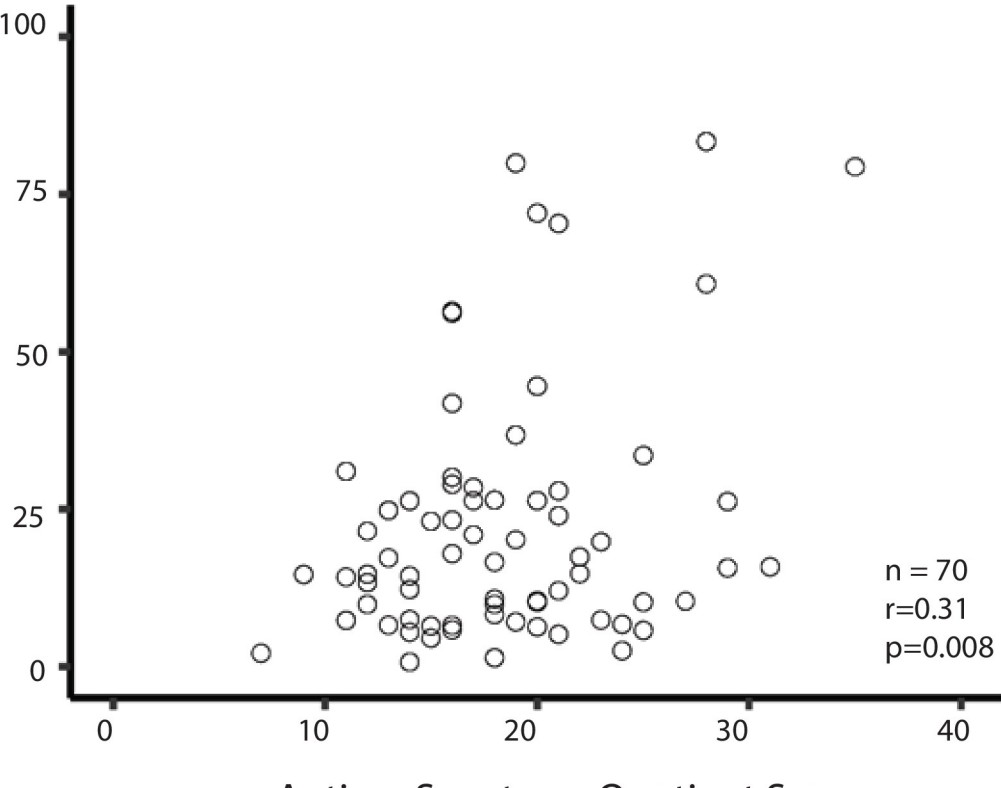

**Fig 5. The relationship between percent of time spent fixating the lower face while viewing sentence videos and Autism-Spectrum Quotient score.** Percent of total fixation time spent fixating the lower face shown on the y axis; AQ score is shown on the x axis. Each point represents one participant.

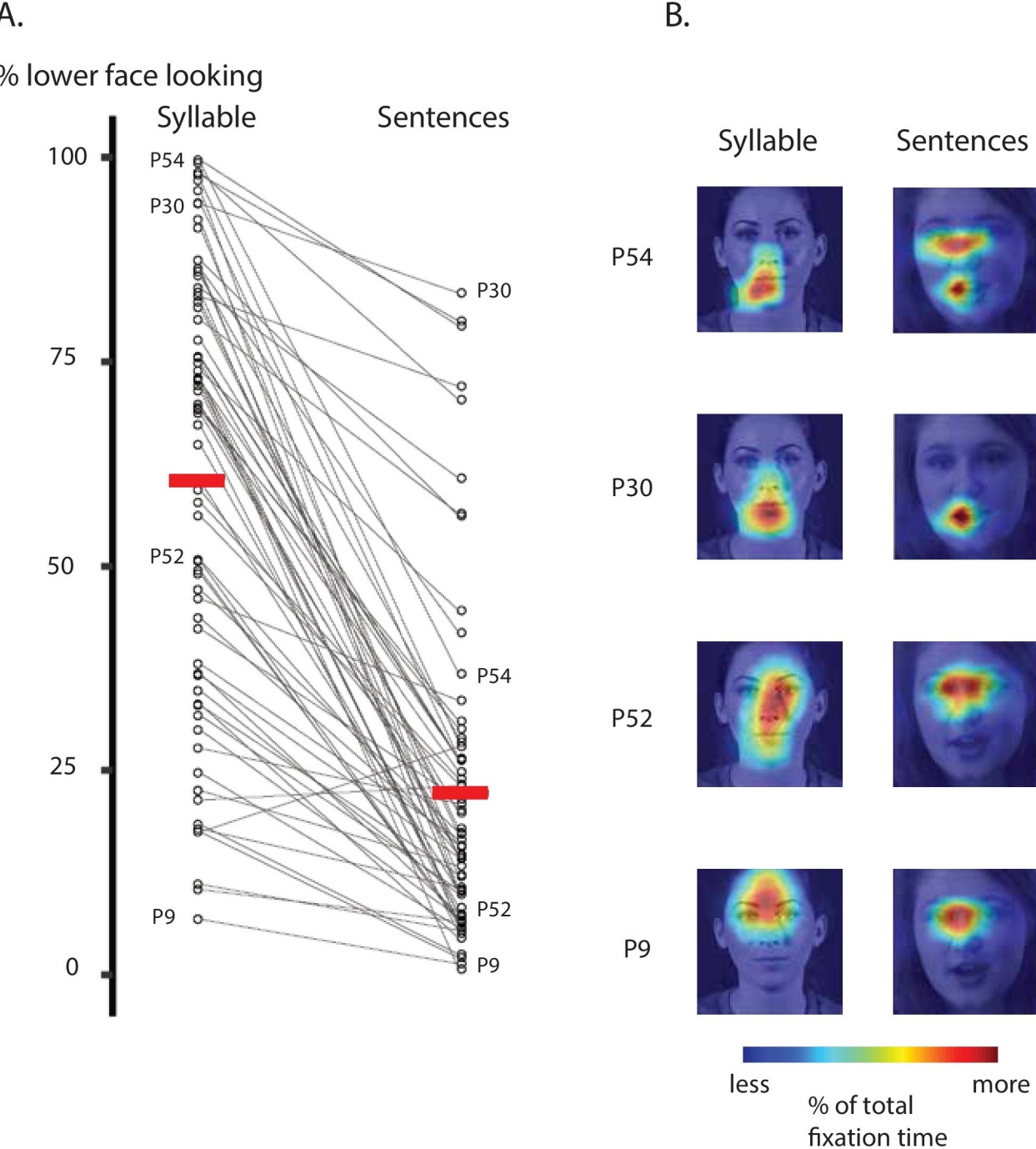

**Fig 6.** A. Comparison of lower face looking by experiment for 70 participants that completed both experiments. The left column represents time spent fixating the lower face during syllable videos, the right column represents time spent fixating the lower face during sentences. Each symbol represents one participant. Symbols for representative participants are labelled (corresponding fixation heat maps are shown in *B*). The mean lower face looking time is shown with a horizontal red line for each experiment. Black lines connect fixation time while viewing syllable movies with the fixation time while viewing sentence movies. Each line represents one participant. B. Fixation heat maps of behavior on each task for four representative participants (labeled by participant number; symbols representing each participant labeled in *A*). Each heat map is visualized on a frame from a single movie, but represent data averaged across all frames and stimuli.

## Experiment 1: Syllable movies

Participants ($n$ = 98; 66 female, mean age 21, age range 18–45) completed a syllable identification task on short audiovisual speech movies. Each trial began with a fixation crosshairs presented outside of the location of where the face would appear in order to simulate natural viewing conditions in which faces rarely appear at the center of gaze [7]. The visual crosshairs then disappeared, and participants were free to fixate. The visual stimulus appeared for 2 seconds, after which the participants had 1 second to report the syllable ("ba,"da," or "ga") they perceived via button press. Four different speakers appeared and repeated each syllable 60 times, for a total of 240 trials. The stimuli were randomized by speaker and syllable, then that same order was presented to all participants. The trials were divided into 4 blocks, separated by rest intervals during which the eye tracker was recalibrated. Total eye tracking time was 12 minutes. Only one speaker was shown per trial and each directly faced the camera. The face subtended approximately 10 cm by 13 cm (6 degrees wide by 8 degrees high).

## Experiment 2: Sentence movies

A subset of the Experiment 1 participants ($n$ = 70, 49 female, mean age 20, range 18–45; all native English speakers) completed a short movie watching task. Participants watched 40 short clips taken from interviews or speeches that had been uploaded to YouTube under the Creative Commons license. As in the syllable videos, each trial began with a fixation cross outside of where the face would later appear. The clips lasted an average of 23 seconds, ranging from 13 and 42 seconds. The clips were shown in approximately 4-minute blocks of 10 videos each, separated by rest intervals where the eye tracker was recalibrated. The total task time was 15.5 minutes. Movies were balanced between blocks by length and speaker gender, then the stimuli were presented in the same order to all participants. Participants were not given a specific task or required to make a behavioral response following the videos in order to encourage as close to natural free viewing as possible. Each movie showed only one speaker and was cropped so that the speaker's face filled most of the frame in order to minimize background distractions. Some clips featured direct gaze from the speakers, while speakers looked to the side in others. A different speaker appeared each trial, with 18 clips featuring female speakers and 22 featuring male speakers. The stimuli filled the entire screen (70 cm x 40 cm) and the face subtended approximately 14 cm by 18 cm (9 degrees wide by 11 degrees high).

## Eye tracking methods and analysis

Participants' eye movements were recorded using an infrared eye tracker (Eye Link 1000 Plus, SR Research Ltd., Ottawa, Ontario, Canada) as they viewed visual stimuli presented on a display (Display++ LCD Monitor, 32" 1920 × 1080, 120 Hz, Cambridge Research Systems, Rochester, UK) and listened to speech through speakers located on either side of the screen. Eye tracking stability was increased with a chin rest placed 90 cm from the display. Eye tracking was performed with a sampling rate of 500 Hz. The eye tracker was calibrated using a 9-target array before each block, for a total of 4 times in each task. Fixations, saccades, and blinks were identified by SR Research's Eyelink software. Saccades, blinks and fixations that started before stimulus onset were excluded from the analysis.

In order to summarize the fixation data, we used a region of interest (ROI) approach. For syllable videos, each fixation in each trial was marked as within or outside the lower face region of interest on each trial. ROIs were hand drawn for each speaker (for sample stimulus and ROI, see Fig 2A). A dividing line was drawn at the tip of the speaker's nose. Since the syllable videos were only 2 seconds and the speaker's head did not move, the same ROI coordinates were used for each frame. Fixations falling in the lower face region were then summed and

divided by the total trial fixation time to calculate a percent time spent fixating lower face on each trial. These trial percentages were then averaged, weighted by the trial duration, to calculate the percent of time spent fixating the lower face for each participant.

The sentence videos were taken from television programs rather than created in the lab, resulting in variation in the face position from frame to frame. Together with the length of the videos, this precluded the use of hand-drawn ROIs. Instead, the Cascade Object Detector in Matlab's Computer Vision System Toolbox was used to automatically generate a box surrounding the face in each video frame. A location at 40% of the face box height was selected as the dividing line before the upper and lower face in order to align most closely to the tip of the nose dividing line used in the syllable task. To check the accuracy of the face detection tool, all ROIs were visualized on top of the original videos. If a face was not identified in a frame or the location was not identified incorrectly, the coordinates from the preceding and succeeding frames were averaged to estimate a location.

The fixation location at the time of presentation of each video frame was measured. To calculate percentage of fixation time, the number of frames with fixation in the lower face were divided by the total number of frames. This value was then averaged across all videos, weighted by the number of frames (since all videos had the same frame rate) in order to calculate a grand mean lower-face fixation percentage for each participant.

## Statistics

A linear mixed-effects model (LME) was constructed with the *lme4* package in R. Statistical values were calculated according to the Satterthwaite approximation using the *lmerTest* package [16]. The model contained a dependent variable of lower-face fixation percentage for each trial, a fixed effect of AQ score, and random effects of participant, stimulus and experimental block (to account for systematic changes in either eye tracker calibration or participant behavior over the course of the experimental session). The participant random effect included both a random intercept (accounting for individual differences in lower-face fixation unrelated to AQ score) and a random slope across experimental blocks (accounting for differential changes in calibration or behavior during the experiment). In the first experiment, the stimulus set contained multiple talkers producing the same set of syllables, so the random effect of stimulus was modeled as syllable nested within talker [% Lower Face ~ AQ + (1| Talker / Syllable) + (1 | Subject) + (0 + Run | Participant)]. In the second experiment, each movie contained a different talker so only a random intercept was modeled [% Lower Face ~ AQ + (1|Stimulus + (1|Participant) + (0 + Run | Participant)].

In addition, a simple linear correlation was calculated with each data point consisting of the lower-face fixation percentage for a participant and the AQ score for the same participant. The $r^2$ value of the correlation provided an estimate of the variance in fixation behavior explained by AQ.

## Heat map visualization

In addition to summarizing the fixation data as a percentage of total fixation time, heat maps were created of all fixation locations. Because different speakers were used, the facial location for each stimulus was aligned so that the heat maps could be collapsed across stimuli.

For the syllable movies, the first frame of each movie was selected. Across these images, the talker's faces were aligned by the tip of their nose. images for the four speakers were then aligned by nose tip and then a 300 x 300-pixel square around this point was selected for each speaker. All fixations with the box for each speaker were selected, then the heat map was constructed by calculating the time spent on each pixel of the image, then dividing by the total

fixation time to create a percentage for each pixel. Doing this allowed a fixation on the nose of one speaker be visualized at the same point on the heat map as a fixation on the nose of a different speaker, even if the actual pixel on the monitor differed.

To visualize average viewing behavior in the sentence task as a single heat map, the first frame of each video was selected as the reference frame and all successive frames were aligned to it. Next, a mean face box was created from the face boxes (600 x 600 pixels) that were automatically identified in each video frame. This mean face box was then used to adjust each original face box to a uniform face box representation. The heat map was constructed by calculating the time spent on each pixel of each image then dividing by the total fixation time to create a percentage of total fixation time for each pixel.

## Results

### Individual differences in face viewing behavior during the syllable videos

In the first experiment, participants were presented with movies of talkers speaking audiovisual syllables (Fig 2A). Participants identified the syllables with high accuracy (mean: 98%). The face was divided into an upper face ROI and a lower face ROI, using the middle of the nose as the dividing line. The eye tracking data was used to calculate the percent of total fixation time spent fixating the lower half of the face for each participant.

Across participants, the mean lower-face percentage fixation time was 62%, but there was substantial variability across participants, ranging from 1% to 99% (Fig 2B). To examine the pattern of face fixations in individual participants, heat maps were generated to show the fixation patterns of three individual participants (Fig 2C) representing patterns of lower (P38), middle (P52), and upper (P9) face looking. Participant 38 exclusively fixated the face below the nose, with most fixation time spent on the mouth and chin and no fixation time spent on the eyes or forehead, corresponding to 99% lower-face looking time. Participant 52 distributed time more evenly across the facial features, reflected by their lower-face percentage fixation time of 50%, with large amounts of time spent on the center of the face in particular between the eyes, on the nose, and on the mouth, and more minimal time spent on the cheeks, forehead, and chin. Participant 9 primarily fixated the speaker's right eye with a moderate amount of time spent on the forehead, nose, and left eyes, and a minimal amount of time spent on the mouth and chin, corresponding to only 7% lower-face fixation time.

### Distribution of Autism-Spectrum Quotient scores

The 98 participants completed the Autism-Spectrum Quotient questionnaire, a self-reported measure of autistic traits with a score range from 0 (no autistic traits) to 50 (high levels of autistic traits). AQ scores varied across from participants from 7 to 35 with a mean value of 19.3 and a standard deviation of 5.7 (Fig 3A).

### Relationship between AQ & face viewing for syllable videos

We constructed a linear mixed-effects model of the relationship between lower-face fixation and AQ that also included random effects of participant, stimulus and experimental block. The LME showed a positive relationship between AQ score and lower-face fixation. Every one-point increase in AQ resulted in 1.2% more time fixating the lower face (p = 0.046, see Table 1 for complete model output). AQ explained 4% of the variance in the inter-participant variability in fixation time (calculated from the $r^2$ of the correlation between AQ and lower-face fixation; Fig 3A).

**Table 1. LME model of lower face looking during Experiment 1 (syllables).**

| Fixed effects | Estimate | Std Error | DF | t-value | p-value |
|---|---|---|---|---|---|
| AQ | 1.175 | 0.580 | 94.97695 | 2.027 | 0.045509 |
| **Random effect** | | **Std. Dev** | | | |
| Participant (slope) | | 8.193 | | | |
| Participant (intercept) | | 26.051 | | | |
| Syllable within talker (intercept) | | 1.363 | | | |
| Talker (intercept) | | 7.236 | | | |

Estimate of the influence of AQ and other factors on % lower-face looking.

## Individual differences in face viewing behavior during the sentence movies

In the second experiment, a subset of 70 participants were presented with longer 13 to 42 second movies of talkers speaking full sentences (Fig 4A). Participants found the sentence movies engaging but did not perform an explicit task. The mean lower-face percentage fixation time while watching the sentence movies was 22%, with substantial variability across participants, ranging from 1% to 83% lower-face fixation percentage (Fig 4B). Fig 4C shows heat maps from participants representing the gamut of fixation patterns, including lower (P30), middle (P13), and upper (P1) face looking (Fig 4C). Participant 30 primarily fixated the mouth with a small portion of time spent on the nose, lower cheeks, and chin (83% lower-face fixation). This behavior resulted in 83% of total fixation time on the lower half of the face. Participant 13 spent a large amount of time on the eyes and the mouth and less on the nose and cheeks (16% lower-face fixation). Participant 1 primarily fixated the speaker's right eye, spent a moderate amount of time on the forehead and left eye, and little to no time on the speaker's nose or mouth (1% lower-face fixation).

## Comparison of syllable and sentence experiments: AQ score

In the subset of 70 participants that completed the sentence experiment, the distribution of AQ scores was similar to the full set of 98 participants (19.3±5.8 for *n* = 98 *vs.* 18.4±5.5 for *n* = 70; p = 0.34).

## Relationship between AQ & face viewing for sentence movies

The LME showed a positive relationship between AQ score and lower-face fixation, with a one-point increase in AQ resulting in 0.8% more time fixating the lower face (p = 0.01, see Table 2 for complete model output). AQ explained 10% of the variance in the inter-participant variability in lower-face fixation time in a simple correlation between the two variables (Fig 5A).

**Table 2. LME model of lower face looking during Experiment 2 (sentences).**

| Fixed effects | Estimate | Std Error | DF | t-value | p-value |
|---|---|---|---|---|---|
| AQ | 0.8436 | 0.3250 | 67.442419 | 2.596 | 0.0116 |
| **Random effect** | | **Std. Dev** | | | |
| Participant (slope) | | 2.702 | | | |
| Participant (intercept) | | 14.408 | | | |
| Sentence movie (intercept) | | 1.730 | | | |

Estimate of the influence of AQ and other factors on % lower-face looking.

## Comparison between the two experiments

There were both differences and similarities between the fixation patterns observed during syllable and sentence viewing. The most important difference was that lower-face looking time decreased from a mean of 61% of total fixation time in the syllable experiment to a mean of 22% in sentence experiment ($t$ = -13.8, $p$ = $10^{-16}$, paired t-test). The effect was consistent, with 68 of 70 participants showing a decrease from the syllable to the sentence experiment (Fig 6A).

The most important similarity was that there was consistency in the participants' preferred fixation locations across experiments as shown by the significant correlation between face viewing during the syllables and sentence movies (r = 0.53, $p$ = $10^{-6}$). To illustrate these effects, fixation heat maps were created for representative participants (Fig 6B). Participant P54 almost always fixated the mouth while viewing syllable movies, but distributed fixations more evenly between the eyes and the mouth while viewing sentences which resulted in an overall decrease in lower face looking (99% to 36%). We observed a similar pattern with another participant (P52), who distributed fixations evenly across facial features while viewing syllable movies but reduced fixations to the nose and mouth when viewing sentence movies (51% to 7%). However, these large shifts were not universal. Some participants maintained the same general behavior across experiments. One participant (P30) fixated the mouth almost constantly on both experiments. This participant did fixate the lower face slightly more on the syllable movies than the (94% to 83%), but in both cases had a much higher than average time spent fixating the lower face. Similarly, a participant (P9) that rarely fixated the mouth during the syllable movies continued to do so in the sentence movies (7% to 1%). Like participant P30, the time spent fixating the lower face did decrease across experiments, but P9 spent much less time than average fixating the lower face in both experiments.

## Discussion

A growing literature demonstrates that healthy participants show large variability in their face viewing behavior across a variety of stimulus and viewing conditions [5,7–12]. These differences have been linked to personality differences and differences in cognitive ability, however, the determinants of this individual variability are not yet fully understood. The motivation for our study was the observation that some healthy controls display face-viewing patterns similar to that observed in individuals with ASD (Fig 1). We sought to determine if autistic traits in healthy controls, as measured with the autism quotient scale, predicted face viewing behavior. We found that they did: a greater tendency to exhibit autistic traits predicted significantly greater mouth-viewing time. This relationship held for both for short videos of talkers enunciating audiovisual syllables and longer videos of talkers speaking complete sentences in a social context.

An aversion to fixating the eyes of a viewed face could arise from the perception of direct gaze as socially threatening [17]. In this account, from an early age, individuals with ASD avoid direct eye contact because it activates neural circuits involved in threat detection [4,18,19]. Greater degrees of autistic traits in healthy individuals might result in direct gaze being perceived as more threatening, resulting in greater avoidance of eye fixations and concomitant negative emotions. A relationship between autistic traits and the degree to which direct gaze is perceived as threatening can account both for some of the group differences in gaze viewing between ASD and controls, and for some individual differences within samples of control participants. Differences in face-viewing behavior might be due to a general relationship that spans both control and individuals with ASD, rather than a specific difference between group of ASD and control individuals. A related concept is the idea of "intolerance of uncertainty" [20], which may mediate the relationship between ASD and anxiety disorders

[21]. Anxiety disorders, including social anxiety, are found in up to 50% of autistic people [22,23]. To the extent that faces evoke anxiety, people with ASD could avoid fixating the eyes (the most relevant part of the face for social communication) or avoid fixating the face entirely. Atypical sensory processing could also play a role in differences in face-looking behavior in people with ASD [24].

## Statistical power and analysis

Three recent meta-analyses have reported that individuals with ASD are less likely to fixate socially relevant stimuli, including the eyes of a viewed face, than healthy controls [2,25,26]. However, all three meta-analyses found a great deal of heterogeneity across published studies, with many published studies finding no significant difference between participants with ASD and controls. Our results suggest that one reason for this heterogeneity may be inadequate power. Of the 132 unique studies included in these meta-analyses, the mean sample size was 20 ASD participants and 21 control participants. If there are large differences *within* groups, as there are known to be for face viewing, then detecting a difference between groups requires a larger sample size than those commonly reported. For instance, to detect a difference between groups of 20% (higher than that reported in the literature) with 80% power, a sample size of 50 participants per group is required [27], larger than nearly all of the studies in the literature. Underpowered studies are problematic in two different ways: they may fail to detect actual differences (type II error) or they may over-estimate the size of the difference due to the existence of the so-called statistical significance filter. If many underpowered studies or statistical comparisons are performed and only the significant comparisons are reported, between-group difference can be inflated by as much as 10-fold [27].

Our study used a dichotomous scheme to classify fixations into either the upper half or the lower half of the face. This was effective because the stimuli consisted almost entirely of a face, with the result that essentially all fixations were to the face (as shown in the individual subject fixation heat maps). For more naturalistic displays containing many objects, it would be important to use a different method for classifying fixations. While feature-specific ROIs are common in the literature, they have a number of drawbacks. First, the experimenter must make an arbitrary decision as to what features constitute an ROI. For example, the peri-mouth region contains visual information about mouth movements due to the structure of facial musculature and could arguably be included in a "mouth ROI". Second, difference ROIs may have unequal sizes, leading to differences in fixation time between ROIs, even under the null hypothesis of random looking behavior. Third, data near the border of the ROI is unnecessarily discarded. For these reasons, unbiased methods that examine the entirety of the fixation data are an attractive possibility for future studies [28].

## Similarities and differences across stimuli

Task instructions can dramatically influence the pattern of eye movements made by participants [6]. Therefore, we presented two very different kinds of stimulus and task configurations. In the syllable experiment, participants reported the syllable spoken by a laboratory-recorded talker during a short 2-second video. In the sentence experiment, participants watched sentences excerpted from videos of conversations without an explicit task. While the syllable videos are tightly controlled, they are also relatively unnatural, with no social or semantic content. The sentence videos are less tightly-controlled, but the presence of social and semantic content make them more similar to the natural conditions encountered in face-to-face interactions [14]. Individual differences were preserved across the syllable and sentence stimuli, as demonstrated by the high correlation in face-viewing behavior across experiments.

However, there were also differences between the experiments, with participants fixating the lower face more for syllables than sentences (68% *vs*. 22%). Participants were required to report the contents of speech only for the syllable videos, and speech tasks drive fixations to the mouth [29]. Stimulus differences also likely contributed to the differences in viewing patterns. The syllable videos were relatively uniform, featuring only 4 speakers, all dressed identically and filmed head-on on an identical background. In contrast, the sentence videos featured 40 different speakers filmed on a variety of backgrounds in a variety of poses. Fixating the eyes is important for face recognition [30,31] and in a recall task, fixations to the eyes increased with the number of different faces viewed [32].

## Other factors contributing to face-viewing variability

AQ scores accounted for a relatively small part of the individual variability in face-viewing, 4% for syllable videos and 10% for sentence movies. Face viewing is a complex human behavior, and it would be surprising indeed if any single factor accounted for a large fraction of individual variation. In addition to AQ, many other aspects of individual differences have been linked to face viewing behavior. While only two fixations are sufficient to identify faces [33], different individuals use different strategies when presented with different face-viewing tasks in an idiosyncratic fashion [34]. Individuals may develop face viewing strategies that maximize their ability to perform the task at hand, and fixate accordingly [9]. A preference for fixating the mouth of a viewed face is correlated with an enhanced ability to understand noisy audiovisual speech [35]. Personality [8] and cultural background [8,36] may also contribute to differences in face viewing behavior.

## Author Contributions

**Conceptualization:** Kira Wegner-Clemens, Johannes Rennig, Michael S. Beauchamp.

**Investigation:** Kira Wegner-Clemens, Johannes Rennig.

**Supervision:** Johannes Rennig, Michael S. Beauchamp.

**Writing – original draft:** Kira Wegner-Clemens, Johannes Rennig, Michael S. Beauchamp.

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
