## [Decision Letter · Decision Letter 0]

30 Oct 2019

PONE-D-19-23722

A relationship between Autism-Spectrum Quotient and face viewing behavior in 98 participants

PLOS ONE

Dear Dr. Beauchamp,

Thank you for submitting your manuscript to PLOS ONE. After careful consideration, we feel that it has merit but does not fully meet PLOS ONE’s publication criteria as it currently stands. Therefore, we invite you to submit a revised version of the manuscript that addresses the points raised during the review process.

Firstly, I must apologise for the delay in getting the reviews to you. It proved quite difficult to find expert reviewers, but thankfully we have. As you will see, both Reviewers see merit in the work and have suggested several areas to improve the manuscript. Specifically, improving the method so that it contains all necessary information to replicate the findings is important. Secondly, I would suggest carefully going through the manuscript to polish the phrasing.

We would appreciate receiving your revised manuscript by Dec 14 2019 11:59PM. To enhance the reproducibility of your results, we recommend that if applicable you deposit your laboratory protocols in protocols.io, where a protocol can be assigned its own identifier (DOI) such that it can be cited independently in the future. For instructions see: http://journals.plos.org/plosone/s/submission-guidelines#loc-laboratory-protocols

We look forward to receiving your revised manuscript.

Kind regards,

Peter James Hills, PhD

Academic Editor

PLOS ONE

Journal Requirements:

3. We note that Figures 1,2,4 and 6includes an image of a [patient / participant / in the study]. 

4.   We note that Figure 4 in your submission contains copyrighted images. All PLOS content is published under the Creative Commons Attribution License (CC BY 4.0), which means that the manuscript, images, and Supporting Information files will be freely available online, and any third party is permitted to access, download, copy, distribute, and use these materials in any way, even commercially, with proper attribution. For more information, see our copyright guidelines: http://journals.plos.org/plosone/s/licenses-and-copyright.

1.    You may seek permission from the original copyright holder of Figure 4 to publish the content specifically under the CC BY 4.0 license.

 In the figure caption of the copyrighted figure, please include the following text: “Reprinted from [ref] undera CC BY license, with permission from [name of publisher], original copyright [original copyright year].”

Reviewers' comments:

Reviewer's Responses to Questions

**Comments to the Author**

1. Is the manuscript technically sound, and do the data support the conclusions?

Reviewer #1: Partly

Reviewer #2: Yes

2. Has the statistical analysis been performed appropriately and rigorously? 

Reviewer #1: I Don't Know

Reviewer #2: Yes

3. Have the authors made all data underlying the findings in their manuscript fully available?

Reviewer #1: Yes

Reviewer #2: No

4. Is the manuscript presented in an intelligible fashion and written in standard English?

Reviewer #1: Yes

Reviewer #2: Yes

5. Review Comments to the Author

Reviewer #1: Two experiments are reported that aim to examine the influence of autistic traits upon face viewing behaviour in a student sample. It was predicted that high levels of autistic traits would be associated with increased fixations towards the mouth region. Experiment 1 involved participants viewing short video clips of a front facing speaker pronouncing a syllable, and then reporting which of three syllables was spoken with a button press. Experiment 2 involved participants free viewing short video clips of a single person saying a full sentence. Simple regressions revealed that AQ score was positively associated with proportion of time spent viewing the lower half of the face, in each experiment, with individuals with higher AQ traits spending longer proportions of time fixating the mouth region.

This work is clearly communicated and directly addresses a research question important for the field of face processing, and has the potential to make a neat contribution to the literature. However, there are aspects of this manuscript that require revision to strengthen the transparency, replicability, conclusions, and rationale (in places). I have listed my comments and suggestions for revision below. Given the data will only be made publicly available after publication, and has not been submitted as part of this review, I have not been able to check analyses.

Introduction

~ASD is not defined. In the Introduction, please define this acronym and provide a brief description of this condition, for non-expert readers.

~The introduction provides a succinct and direct rationale for examining the influence autistic traits have upon face viewing behaviour. However, I would suggest you acknowledge the prior research studies that have also examined the influence of autistic traits upon face viewing behaviour, for balance and completeness. One example of this work is Vabalas and Freeth (2015, Journal of Autism and Developmental Disorders).

~The Introduction provides a rationale for the influence of autistic traits upon face viewing; however, no rationale or justification is presented for why or how this may differ across viewing conditions and why two experiments were conducted using different types of stimuli. This needs to be addressed, perhaps either in the Introduction, or in a Discussion section that separates the Methods/Results of the two Experiments.

Method

~The replicability of the Method could be improved by including more detail e.g., accuracy of calibrations, detail on any counterbalancing and/or randomisation, software used for data presentation/extraction. Please also double check the trial numbers/eye tracking time reported – these don’t quite add up based on my own calculations – perhaps it is the explanation of this that needs clarifying? Making these alternations would not only mean the Method could be directly replicated, but also allow for the quality of data to be evaluated by readers.

~Related to the above point, the Method is missing a few of the standardised sections one would expect to be included (e.g., Design and Participants). This made finding this information somewhat lengthy and more challenging – I would strongly advise you consider including these sections and adding in additional detail where necessary.

~Page 8 line 157: For Experiment 1 it is stated that the proportion of fixations included upon the lower half of the face were weighted by trial duration – this seems to contradict earlier information that all trials were 2 seconds in duration. I would advise this be clarified.

~Page 9 line 180: I have noticed a typo - “noise”

Results

~The field is now adopting the use of linear mixed effect models, which avoid issues with averaging across trials, and provides a more sensitive analysis, accounting for random variance associated with both participants and stimuli. I would suggest the authors consider adopting this technique to analyse the current data. I understand this is not a trivial task, however, there are now an abundance of very helpful and user friendly guides and information on how to conduct and interpret mixed model analysis.

~The analysis of the data focused upon examining proportion of fixation time spend within ROI subsuming the lower half of the face. In places, the authors appear to assume this means the remaining fixations were directed towards the upper half of the face (e.g., eye region). I accept that this is very likely, however, there is evidence that attention is directed to non-social information (e.g., background), when viewing social scenes (e.g., images that contain a face or person), in samples of participants with a diagnosis of autism. Therefore, I would advise the authors report data on fixations upon the upper half of the face too, if they are to conclude autistic traits are associated with reduced gaze towards this region, in addition to increased fixations upon the lower face/mouth region. This is critical for the interpretation of this data.

~I was surprised to read the effect of gender on viewing behaviour was analysed. What was the rationale for conducting this analysis? If this is exploratory, I would suggest it is important for the authors to explicitly state this. In addition, given no influence of gender was detected in the regression analysis, it is unclear why post hoc analyses were conducted.

Discussion

~Throughout the report there are some very strong claims about the lack of understanding of the mechanisms that underpin individual variability in face viewing behaviour (e.g., page 14 line 293 – “the determinants of individual variability are unknown”). I question whether this accurately reflects the literature… I would suggest the authors might want to consider lessening the strength of these claims to acknowledge that some understanding of individual variation in face viewing behaviour has been developed, for example, the impact of anxiety and cultural background.

~There are a couple of sentences that could be clearer, so the authors may want to revisit the manuscript to increase precision in places e.g., “Our results demonstrate that individual’s face viewing behaviour can be predicted without actually measuring it…” Given eye tracking is a direct measure of face viewing behaviour, I am not sure what the intended meaning of this sentence is.

~Page 15 line 303. The authors provide a fair explanation for the possible mechanisms that underpin autistic traits and the differential face viewing behaviour. I would suggest that for a more complete and balanced view the authors should also acknowledge the range of alternative explanations too.

~In the Discussion, the authors make a good and valid point about the statistical power of previous studies. However, in the report there is no report of an a priori power analysis conducted for the current experiments. To strengthen this point and the manuscript, if a priori power analysis this was completed and contributed to the design of the current experiments (which the manuscript implies), this should be reported in the Method.

Reviewer #2: SUMMARY

Many studies have suggested that autistic individuals present atypical looking patterns to faces. This study examines whether similar differences could be found in the general population when considering individual differences with respect to autism traits. Using AQ to quantify autism traits, this eye-tracking study provides evidence that individuals with higher levels of autism traits tend to look more in the mouth area, in two types of short video stimuli.

Comments

This is a very interesting paper, which is also presented in a clear and engaging way. The question of whether individual differences in the levels of autism traits could account for differences in the way we scan or process faces is an important one. The study employs a large sample and powerful methods to ask this question.

I would recommend accepting this study for publication, subject to addressing minor issues, which are however important for presenting the findings in a convincing way.

P.6, 116 - The gender distribution is not balanced. What are the implications of this imbalance for the analysis of gender effects?

P.6, 118 - It would be good to provide additional details on the experimental setup. How far on average from the fixation crosshair did the stimuli appear? I assume stimuli have been always presented in the middle of the screen?

P.8, 155-159 - Why was a more precise analysis of looking preferences not adopted? Could the authors please justify their choice of analysis? I would anticipate that the measure to correlate with AQ should be the looking preferences towards the eyes. This would establish consistency with the autism literature and would also allow aligning the assessment of looking preferences in the two types of stimulus (short vs longer videos). I think that dichotomising the screen provides a rather coarse measure of looking strategies. What if people have looked at the corners of the screen or at the clothes? I also wonder if the use of this measure in this study underlies the difference between the two experiments shown in Figure 6. Have the authors considered this possibility?

P. 9, 166-168 - How often did this happen?

P. 9, 176-186 - The heatmap visualisation method is self-explanatory, however, it is only applied at an individual level. It is difficult to know how representative is the example heatmaps provided for some of the participants. Could the authors provide a super-subject heatmap visualisation or alternatively make all the individual heatmaps available?

The main concern is again the dichotomous measure of looking preferences in the first experiment, which does not convince me that all participants looked at faces only.

P.11, 223-226 - So is the correlation with looking at the upper part of the screen -0.19?

P.11, 230, Not sure why the engagement of participants was concluded here.

P.12, 249-251 - So is the correlation with looking at the eyes -0.31. If not, is this difference from Experiment 1 responsible for the difference discussed in Figure 6?

P.14, 279-288 - see earlier comment given the gender imbalance in the sample.

P.15, 303-305. Have the authors examined their data considering subscales of AQ? Should the correlation be higher for the social subscale under this account?

Discussion. Some very interesting points were raised, however, I would welcome discussing this study more broadly. Could other constructs relevant to autism-like sensory sensitivities or intolerance to uncertainty also account for some variance? What would be the next steps for this very interesting study?j

I could not locate the data in the Dryad repository.

6. PLOS authors have the option to publish the peer review history of their article (what does this mean?). If published, this will include your full peer review and any attached files.

Reviewer #1: No

Reviewer #2: No

---

## [Author Response · Author response to Decision Letter 0]

22 Jan 2020

Please see the uploaded "Response to Reviewers" file for complete response.

---

## [Decision Letter · Decision Letter 1]

25 Feb 2020

PONE-D-19-23722R1

A relationship between Autism-Spectrum Quotient and face viewing behavior in 98 participants

PLOS ONE

Dear Dr. Beauchamp,

Thank you for submitting your manuscript to PLOS ONE. After careful consideration, we feel that it has merit but does not fully meet PLOS ONE’s publication criteria as it currently stands. Therefore, we invite you to submit a revised version of the manuscript that addresses the points raised during the review process.

Please respond to the clarification points made by Reviewer 1. These are minor tidying up issues that will help provide balance. I am happy for both analyses to remain in the manuscript if you prefer, however, I leave the final decision up to you.

We would appreciate receiving your revised manuscript by Apr 10 2020 11:59PM. To enhance the reproducibility of your results, we recommend that if applicable you deposit your laboratory protocols in protocols.io, where a protocol can be assigned its own identifier (DOI) such that it can be cited independently in the future. For instructions see: http://journals.plos.org/plosone/s/submission-guidelines#loc-laboratory-protocols

We look forward to receiving your revised manuscript.

Kind regards,

Peter James Hills, PhD

Academic Editor

PLOS ONE

Reviewers' comments:

Reviewer's Responses to Questions

**Comments to the Author**

1. If the authors have adequately addressed your comments raised in a previous round of review and you feel that this manuscript is now acceptable for publication, you may indicate that here to bypass the “Comments to the Author” section, enter your conflict of interest statement in the “Confidential to Editor” section, and submit your "Accept" recommendation.

Reviewer #1: (No Response)

Reviewer #2: All comments have been addressed

2. Is the manuscript technically sound, and do the data support the conclusions?

Reviewer #1: (No Response)

Reviewer #2: Yes

3. Has the statistical analysis been performed appropriately and rigorously? 

Reviewer #1: (No Response)

Reviewer #2: Yes

4. Have the authors made all data underlying the findings in their manuscript fully available?

Reviewer #1: (No Response)

Reviewer #2: Yes

5. Is the manuscript presented in an intelligible fashion and written in standard English?

Reviewer #1: (No Response)

Reviewer #2: Yes

6. Review Comments to the Author

Reviewer #1: I thank the authors for responded to each of the reviewer comments. I have made some minor suggestions below that I would suggest are necessary before publication (following the response to reviewer comments).

1. It is great to see you have made raw data available, however, this would benefit from the inclusion of a meta-data file to explain what file and column represents. If you have any analysis scripts these would be useful to make available too, so readers can reproduce your analysis.

2. I suggested the authors consider using mixed effects models to analyse their data – this has been done, however, has been included as additional analyses as opposed to replacing the simple regression. The authors should choose one or the other technique to report. If you choose to include the mixed effects models, these should be reported in full. One reason perhaps against reporting the linear mixed effects models, is that I assume you conducted the power analysis for a simple regression? I apologise for this contradicting my initial suggestion, but this has only become apparent since your response to the original reviews re the a priori power analysis you conducted.

3. I accept the authors response re the decision to focus analysis on only the lower half of the face. However, an explanation within the paper for this decision and evidence to support this decision is needed in the paper, else readers will likely question this.

4. It was surprising that the authors included reference to cultural differences in eye movement patterns during face processing, given they state they could not find evidence for this. There is research demonstrating these differences exist – if the authors choose to include reference to cultural differences, it would be appropriate to also refer to this research for balance. (e.g., see Blais et al., 2008)

5. Both reviewers suggested a more in-depth discussion of the range of theoretical reasons for autistic traits influencing face scanning be included in the Discussion (Reviewer 1 point 12, Reviewer 2 point 12) – this point does not appear to have been addressed – the authors might want to consider including some discussion on this topic.

Reviewer #2: The authors have addressed all my comments. A minor issue is that a sentence in the response to point 6 was incomplete. However, I am happy to take the authors point about Mehoudar, et al. (2014) on board.

I would suggest that the revised paper should be accepted for publication in PLOS ONE.

7. PLOS authors have the option to publish the peer review history of their article (what does this mean?). If published, this will include your full peer review and any attached files.

Reviewer #1: No

Reviewer #2: No

---

## [Author Response · Author response to Decision Letter 1]

4 Mar 2020

Please see the uploaded "Response to Reviewers" file.

---

## [Editor Report · Decision Letter 2]

11 Mar 2020

A relationship between Autism-Spectrum Quotient and face viewing behavior in 98 participants

PONE-D-19-23722R2

Dear Dr. Beauchamp,

We are pleased to inform you that your manuscript has been judged scientifically suitable for publication and will be formally accepted for publication once it complies with all outstanding technical requirements.

With kind regards,

Peter James Hills, PhD

Academic Editor

PLOS ONE
---

## [Editor Report · Acceptance letter]

20 Apr 2020

PONE-D-19-23722R2 

A relationship between Autism-Spectrum Quotient and face viewing behavior in 98 participants 

Dear Dr. Beauchamp:

I am pleased to inform you that your manuscript has been deemed suitable for publication in PLOS ONE. Congratulations! Your manuscript is now with our production department. 

With kind regards,

on behalf of

Dr Peter James Hills 

Academic Editor

PLOS ONE